# A Comprehensive Review on Natural Fibers: Technological and Socio-Economical Aspects

**DOI:** 10.3390/polym13244280

**Published:** 2021-12-07

**Authors:** Azizatul Karimah, Muhammad Rasyidur Ridho, Sasa Sofyan Munawar, Yusup Amin, Ratih Damayanti, Muhammad Adly Rahandi Lubis, Asri Peni Wulandari, Apri Heri Iswanto, Ahmad Fudholi, Mochamad Asrofi, Euis Saedah, Nasmi Herlina Sari, Bayu Rizky Pratama, Widya Fatriasari, Deded Sarip Nawawi, Sanjay Mavinkere Rangappa, Suchart Siengchin

**Affiliations:** 1Research Center for Biomaterials, National Research and Innovation Agency (BRIN), Jl Raya Bogor KM 46, Cibinong 16911, Indonesia; karimahazizatul@gmail.com (A.K.); rasyidmuhammad0505@gmail.com (M.R.R.); sasa001@brin.go.id (S.S.M.); isma011@brin.go.id (I.); yusu007@brin.go.id (Y.A.); muha142@brin.go.id (M.A.R.L.); 2Forest Products Research and Development Center, Ministry of Environment and Forestry, Bogor 16610, Indonesia; ratih_turmuzi@yahoo.com; 3Department of Biology, Faculty of Mathematics and Science, University of Padjajaran, Jatinangor 45363, Indonesia; asri.peni@unpad.ac.id; 4Indonesian Sweetener and Fiber Crops Research Institute (ISFCRI), Ministry of Agriculture, Malang 65152, Indonesia; nurarindatta@gmail.com; 5Department of Forest Product, Faculty of Forestry, Universitas Sumatera Utara, Medan 20155, Indonesia; 6JATI-Sumatran Forestry Analysis Study Center, Jl. Tridarma Ujung No. 1, Kampus USU, Medan 20155, Indonesia; 7Solar Energy Research Institute, Universiti Kebangsaan Malaysia, Bangi 43600, Malaysia; a.fudholi@ukm.edu.my; 8Research Centre for Electrical Power and Mechatronics, National Research and Innovation Agency (BRIN), Kawasan LIPI Cisitu, Bandung 40135, Indonesia; 9Department of Mechanical Engineering, Kampus Tegalboto, University of Jember, Jember 68121, Indonesia; asrofi.net@gmail.com; 10Center for Development of Advanced Science and Technology (CDAST), Advanced Materials Research Group, Kampus Tegalboto, University of Jember, Jember 68121, Indonesia; 11Indonesia Natural Fiber Council (DSI), Gedung Smesco/SME Tower Lt. G (APINDO UMKM Hub), Jl Gatot Subroto Kav. 94 Pancoran, Jakarta Selatan 12780, Indonesia; euis@inafiber.id; 12Department of Mechanical Engineering, Faculty of Engineering, University of Mataram, Mataram 001016, Indonesia; n.herlinasari@unram.ac.id; 13The Graduate School, Kasetsart University, Chatuchak, Bangkok 10903, Thailand; bayu_rizkypratama@yahoo.com; 14Department of Forest Products, Faculty of Forestry and Environment, IPB University, Bogor 16680, Indonesia; 15Natural Composites Research Group Lab, Department of Materials and Production Engineering, The Sirindhorn International Thai-German Graduate School of Engineering (TGGS), King Mongkut’s University of Technology North Bangkok, Bangkok 10800, Thailand; suchart.s.pe@tggs-bangkok.org

**Keywords:** natural fibers, socio-economic assessment, technological aspects, sustainability, renewable resources

## Abstract

Asian countries have abundant resources of natural fibers, but unfortunately, they have not been optimally utilized. The facts showed that from 2014 to 2020, there was a shortfall in meeting national demand of over USD 2.75 million per year. Therefore, in order to develop the utilization and improve the economic potential as well as the sustainability of natural fibers, a comprehensive review is required. The study aimed to demonstrate the availability, technological processing, and socio-economical aspects of natural fibers. Although many studies have been conducted on this material, it is necessary to revisit their potential from those perspectives to maximize their use. The renewability and biodegradability of natural fiber are part of the fascinating properties that lead to their prospective use in automotive, aerospace industries, structural and building constructions, bio packaging, textiles, biomedical applications, and military vehicles. To increase the range of applications, relevant technologies in conjunction with social approaches are very important. Hence, in the future, the utilization can be expanded in many fields by considering the basic characteristics and appropriate technologies of the natural fibers. Selecting the most prospective natural fiber for creating national products can be assisted by providing an integrated management system from a digitalized information on potential and related technological approaches. To make it happens, collaborations between stakeholders from the national R&D agency, the government as policy maker, and academic institutions to develop national bioproducts based on domestic innovation in order to move the circular economy forward are essential.

## 1. Introduction

Scientists, researchers, and practitioners around the world have recently been working to maximize the potential of natural fibers to create the most sustainable, biodegradable, and high-quality natural fiber products [1,2,3]. Natural fibers, which are renewable and ecologically acceptable sources of raw materials for producing environmentally friendly products, have played a significant part in human civilization [1]. Natural fibers have many advantages over synthetic fibers, including lower density, lighter weight, lower cost, biodegradability, minimal health hazards during processing, abundant availability and ease of availability, low investment at low cost for production, lower energy requirements, and lower CO_2_ emissions, indicating that they have great potential as a substitute for glass, carbon, or other synthetic fibers. Natural fibers are desirable bio-sourced materials as an alternative to non-sustainable glass and carbon fiber reinforced composites owing to their availability and technical viability.

From the physical and mechanical properties point of view, natural fiber has relatively high tensile strength and Young’s modulus, good thermal, good acoustic insulation characteristics, and high electrical resistant [1,2,3,4,5,6,7,8]. Furthermore, chemical properties of natural fibers, such as high cellulose content, have a strong relationship with tensile properties, crystallinity, and density [1,7]. Notwithstanding, natural fibers have some drawbacks that need to be enhanced, such as low impact strength, non-uniformity in quality and price, poor moisture resistance, low durability, low compatibility, low adhesion efficiency, moisture absorption, and poor wettability [9,10,11,12,13,14]. Therefore, to achieve adequate uses and overcome some natural fiber limitations such as biocompatibility and hydrophilic properties, appropriate technologies should be applied for instance by surface modifications and chemical treatment methods [3].

Natural fibers have been successfully applied to a wide range of applications, for instance, furniture, automotive, electronic industries, and building construction. According to Ahmed et al., the wear resistance of *Areva javanica* fiber brake pads is 16% higher than acrylic fiber-based brake pad; hence, the *A. javanica* fiber can be used as a possible substitute for synthetic acrylic fiber-based disc brake pads [15]. One example in the automotive sector is the utilization of hybrid kenaf-glass reinforced epoxy composite for passenger car bumper beams [16]. Chandramohan and Bharanichandar [17] also develop natural fiber reinforced polymer composites for automobile accessories and conclude that one of the best materials is the hybrid of sisal and rosella. Plastic/wood fiber composites are used in a variety of applications, including decks, docks, window frames, and molded panel components [18]. Furthermore, corn husk fiber/polyester composites have also been used as sound absorbers to replace glass fiber composites [19].

According to all the development technologies and the availability of natural fibers mentioned above, the utilization of natural fibers can improve economic growth and the well-being of citizens. Industries that use natural fibers as a raw material provide a major source of revenue. Various studies have already been conducted on the use of natural fibers, namely, as reinforced composites in biomedical applications [20,21,22], automotive devices [23], aerospace [24], and textile resources [25]. This study provides more information about the potential of Indonesian natural fibers from technological and socio-economical aspects.

## 2. Overview on Natural Fibers

The most common classification for natural fibers is from botanical forms. Natural fibers can be classified into five types [26]: Other forms include (1) bast fibers (for example abaca, sisal, pineapple), (2) leaf fibers (such as ramie, flax, kenaf), (3) seed fibers (coir, cotton, and kapok), (4) grass and reeds (wheat, corn, and rice), and (5) wood and roots. A more comprehensive list of fiber classifications can be found in Figure 1. A variety of fibers are produced by several plants. For example, jute, flax, hemp, and kenaf have both bast and core fibers, whereas agave, coconut, and oil palm have both fruit and stem fibers. Both stem and hull fibers can be found in cereal grains [27].

The musa plants (*Musa acuminata*) are native to the South-East Asia and belong to the Musaceae family [6,28,29]. This plant produces biomasses that are categorized as useful materials with high fiber materials, such as bunches, pseudo-stems, leaves, and stalk [30]. Banana is widely available in tropical countries such as Malaysia and South India [31]. It is the fourth most important crop in developing countries [32]; meanwhile, tropical and subtropical countries also have sufficient natural resources [6,28,29]. However, approximately 88.84% of the waste with a high fiber content was discarded [33]. The tree becomes waste after one season of fruit harvesting and cutting it allows for the growth of new plants [34]. After the banana trees have been cut down, they are dried and processed to extract the fiber [34,35]. High low elongation at break, light weight, good fire resistance, strong moisture absorption, low density, high tensile strength, and modulus are some of the advantages of banana fiber [36].

Abaca (*Musa textilis*) fiber is classified as leaf fiber in some classifications, but it is classified as a stem fiber in others, especially those derived from pseudo stem [37]. This plant grows to a height of 4.5 to 7.5 m and has a stem diameter of 12.5 to 20 cm. The stem was surrounded by a tangle of large, piled-up fronds. Fibers with a width and length of 12.5 to 30 cm and 1.5 to 2.5 m, respectively, are found in the leaf fronds [38]. The abaca fiber has high tensile strength and is resistant to banding and sea water. The position of the frond on the stem influences its fiber quality, with the outer part being stronger than the inner part. Additionally, abaca fiber has excellent flotation properties [38]. Abaca is widely planted in Talaud, North Sulawesi, and from this location, five local superior varieties were released in October 2019 [39].

Pineapple leaf fibers (*Ananas comosus* L. Merr.) are one of the waste materials available in South East Asia, India, and South America that has not been fully explored [40]. In addition, after Philippines and Thailand, Indonesia contributes 23% of pineapple production. Almost all of Indonesia’s regions have ideal tropical climates for growing pineapple plants. On the other hand, the pineapple plantation needs more development by applying superior varieties and appropriate cultivation techniques [41]. These plants grow in moist and dry climates, as well as tropical and subtropical regions [42]. However, these plants tend to grow at temperatures between 18–45 °C and at altitude below 800 mdpl, due to high altitude and extreme temperatures influencing the size and quality of pineapple plants [40]. With a fiber yield of 1.55% to 2.5% [43], pineapple leaf fibers could become a new source of raw material for industries [40], such as polymer composites reinforcement [42], and could also be used to replace synthetic fibers [40,43]. Additionally, they have a softer texture than other vegetable fibers [43], as well as a high strength and smooth surface [42].

Bamboo is a type of lignocellulose material from the grasses (Graminae) family that has a wide range application as a potential fiber source. Bamboo fiber is a type of natural fiber that aligns in longitudinal directions, according to Zakikhani et al. [44] and Wang and Chen [45]. After India and China, Indonesia is ranked the third place in bamboo production [46]. Bamboo has the highest productivity [47], is easy to grow, and harvests quickly when compared to other non-wood forest products [48,49]. Indonesia has 160 bamboo species, 88 of which are endemic [50]. Betung bamboo (*Dendrocalamus asper* (Schult.f)) is one of the most common species in Indonesia [51]. Betung bamboo has better fiber morphology and physical–chemical properties compared to the other species, followed by the yellow bamboo (*Bambusa vulgaris* Scharader ex Wendland), andong bamboo (*Gigantochloa pseudoarundinaceae* (Steud) Widjaja), tali bamboo (*Gigantochloa apus* (Schutz)), and black bamboo (*Gigantochloa atroviolacea* Widjaja) [52]. In the textile industry, this plant is used in two ways: to produce natural (bast) fiber by physical and chemical treatment and to spin regenerated (pulp) fiber through retting bamboo plant into pulp [53].

Cotton fiber (*Gossypium* sp.) is a type of fruit fiber that is used as a primary raw material for textile, health, and beauty products [54]. The cotton production was about 2.558 tons, with a plantation area of 11.287 Ha [55]. Even though the domestic cotton demand is increasing, the supply is not keeping up. Low cotton production and less farmer interest in planting are to blame [56]. Cotton is classified into three classes based on the length and smoothness of the fibers: long, medium, and short cotton fibers with lengths of 1.5, 0.5 to 1.3, and 0.3 to 1 inch, respectively [20]. Training on cultivation techniques and the use of superior seeds are needed to increase cotton productivity [57].

Ramie (*Boehmeria nivea* S. Gaud) is a kind of compatible fiber, making it is simple to be combined with a variety of other fibers [25]. Ramie is a fast-grown and branchless plant that can reach a height of 1 to 2 m. The stem-extracted bast fibers are the strongest and the longest natural bast fibers [58]. The productivity of ramie fiber is determined by the stem’s height, diameter, skin thickness, and fiber yield (fiber content per stem). Since the fiber production can be carried out every 2 months, harvesting is conducted 5 to 6 times per year in tropical areas like Indonesia. China grass crude fiber contains around 2–4% of fresh stalk, 1–3% of degummed fibers, and 1–2% of hemp top [59,60]. Furthermore, ramie production is about 100 thousand tons per year, which is higher than abaca fiber production, at about 70 thousand tons per year [61]. According to Soeroto [62], these plants could grow in Indonesia’s middle to highland areas, with the highest productivity in the highlands (700 mdpl) from 2.5 to 3.0 tones/ha/year [63]. Ramie productivity per hectare is much higher than cotton [64] and its fiber quality is higher than cotton with the color and luster comparable to natural silk. Furthermore, ramie absorbs 12% of water while cotton absorbs just 8% [65].

Sisal (*Agave sisalana* L.) is a good fiber-producing plant that can be grown on dry land and is resistant to soil with a high salt content [66]. Because of its ease of cultivation and fast renewal times, sisal fiber accounts for half of all textile production [67]. Indonesia produces 500 tons of sisal fiber per year, which is obtained from the plant’s leaves [61]. According to Mukherjee and Satyanarayana [67], sisal plants can produce 200 to 250 leaves per year, with each leaf containing 1000 to 1200 fiber bundles. Furthermore, each bundle contains 87.25% water, 4% fiber, 0.75% cuticle, and 8% dry matter. Sisal fibers are characterized by their hardness, strength, and yellowish white color. Each leaf of 1000 fiber bundles contains 4% fiber [68], which has not been used to its full potential [37].

Coconut (*Cocos nucifera*) is a plantation plant that produces fiber from its fruit for use in furniture, crafts, and probably polymer composite reinforcements [69]. Coir fiber can be obtained from coconut, and it has the thickest, the most resistant, and the lowest decomposition rate of all-natural fibers. It is ideal for rope production due to its high strength [70]. Fruit cultivation and pruning produce a significant amount of coir fiber [71,72]. Furthermore, coconut cultivation produces coir and pith, which account for around 35% of the total weight of the crop [73].

Sansevieria is a genus of ornamental plants in the Agavaceae family that grows from lowlands to highlands in the tropics and sub tropics [74]. This plant is xerofit with thick leaves due to high moisture content [75] with a spherical, half-shaped leaf style round, stiff as a blade, short curved, and sunken fleshy. The leaves have smooth and corrugated margins, and the tips are tapered, pointy, and blunt [76]. In addition, fibers from Sansevieria were extracted using the retting method [14]. It may be used as textile raw materials, an absorbent of pollutants, and cancer cell inhibitor [76,77,78,79,80]. It is also used to treat diseases like stomach pain, earaches, diarrhea, hemorrhoids, fungi, scabies infections [81], as well as for anti-inflammatories, analgesics, antipyretics, antioxidants, and antimicrobial activity [82], and as raw material for handicrafts [83].

Jute fibers (*Corchorus capsularis* and *C. olitorius*) are off-white to brown in color and range in length from 1 to 4 m, which is obtained from the bast or skin of the plant. Jute fibers with a large amount of cellulose, high tensile strength, and low extensibility could be grown in 4 to 6 months. They have better fabric breathability, are free of narcotics or odor, have strong insulating and anti-static properties, low thermal conductivity, and mild moisture recovery. Jute fibers are appealing because they are biodegradable, recyclable and environmentally friendly [84]. According to Suliyanthini [38], these fibers have very low creep, are brittle, and have a coarse nature that limits the fineness of the yarn. Packaging, sack material, tapestry coatings, electrical insulation, an industrial fiber for carpet coatings, electrical, rigging, tarpaulin, roofing materials, automotive manufacture, and straps are some of these fiber applications [38,85]. Because of broken hair that may cause food contamination, jute fiber is not appropriate for certain forms of food [38]. Jute plants have short, tall, straight stems with leaves at the top of the tree, and the fibers are derived from them. The jute tree grows to a height of 1.5 to 4.8 m and has a stem diameter of 1.25 to 2.0 cm [38]. Furthermore, the retting method may be used to draw these fibers [86].

Kenaf fiber comes from the *Hibiscus cannabinus* L. plant stem, which has been grown since 1979/1980 as part of the ISKARA (intensification of community sack community) program [87]. These plants are adaptable and can be grown on a variety of surfaces, including peat [88] and flooded soil [88,89]. Depending on the variety and growing climate, kenaf productivity can range from 2.0 to 4.0 tons of dry fiber/ha [90]. It is an annual plant with a stem diameter of 1.25 cm and a height of 2.5 to 3.75 m [38]. According to previous records, India and Pakistan are the world’s top kenaf producers. Tropical and subtropical climates with high humidity, heavy rain, and no strong winds are ideal for these plants. They thrive in loose, well-draining soil and are planted similarly to jute. They can be harvested 4 to 5 months after they begin to bloom [38,91].

*Bombix mori* caterpillar cocoons are used to make silk fiber [35,92]. Due to its high tensile strength, strong degree of resilience, elasticity, flexibility, biodegradation, and great biocompatibility, it is considered a possible biomaterial that supports cell attachment and proliferation [92,93,94,95,96,97,98]. Silk fibers are strong, smooth and crease resistant, with a high capacity to absorb water. Furthermore, these fibers are used in the manufacture of women’s clothes, socks, ties, and tissue engineering [35]. They are considered as the most desirable and coveted fibers because of their relative rarity, unique luster, softness, and drape [99]. Wool is the most essential animal fiber, and it is obtained from sheep, in either a staple or short form. It contains keratin protein, lanolin (an external lipid), and a small amount of internal wool lipid (about 1.5%) [100,101]. Clothing, sweaters, blankets, rugs, weaving, and knitting all use wool as a raw material [35].

Collagen is a connective tissue extracellular matrix that is derived from the skin and bones of animals and comprises 30% protein [102]. It is commonly used in biomedicine, medicinal food, food and drug growth, and cosmetics [103]. It has been used as a homeostatic agent, bone tissue regeneration, membrane oxygenator, contraception (barrier method), implant, and drug delivery system in biomedicine. In the cosmetics industry, collagen is used as an emulsifier and foaming agent in the food industry, while in the field of cosmetics, it becomes an active ingredient used to avoid the incidence of premature aging (anti-aging) [103]. Chemical processes and the combination of both enzymatic and chemical processes [104] have been used to isolate collagen (acid-soluble collagen and pepsin-soluble collagen) [105]. To make certain products, collagen fibers in animal skin are processed into leather through a tanning process [103]. Footwear, clothing, gloves, leather goods, heavy leather, and upholstery are all made of leather [35].

Corn is the most common crop found in every region of many Asian countries. The potential of corn plants in providing natural fiber is very high, such as the stems, leaves, and skins of corn. Cornhusk fiber contains cellulose, hemicellulose, and lignin of 46.15%, 33.79%, and 8.92%, respectively. The tensile strength value of corn husk fiber is 169.49 MPa, which is higher than the tensile strength of glass fiber, which is 1.7–3.5 MPa [7]. Modification of corn husk fiber using sodium hydroxide (NaOH) solution with a concentration of 0.5–8% is known to reduce the hydrophilic properties, and increase the crystallinity, tensile strength, and thermal resistance of the fibers [7]. The addition of corn husk fiber to polymer composites can increase the tensile strength, bending strength, and toughness properties of the polyester composite [19,106]. Although immersed in water and exposed to ultraviolet (UV) light, the mechanical properties of the corn husk fiber composite were still quite high compared to the “pandan wangi” fiber-reinforced composite. In several previous studies, this corn husk fiber composite was found suitable to be used as a substitute for wood, soundproofing panels, and building materials [19,106]. Figure 2 shows some natural fiber resources.

## 3. Technological Perspective of Natural Fibers Processing

Temperature, humidity, height, growing site, local climatic conditions, season, and harvesting are all factors influencing the quality and structure of natural fibers [39,107,108,109]. Handling method, storage period and condition, and harvested plant portion can all affect fiber qualities [107,108,109]. Some of the elements should be closely monitored to acquire the best fiber characteristics.

Natural fibers include kapok, ramie, pineapple, sansevieria, kenaf, abaca, sisal, and coconut fiber, as well as bamboo [110]. Bamboo has a greater ultimate strength [111] than other fiber bioresources like jowar and sisal, which may be tested in single unit fiber or fiber-bundle tests. Practically, the last test is preferable because it is easier to administer and yields faster findings [110]. The mechanical properties of fibers were also influenced by their microstructure and chemical composition (cellulose, hemicellulose, and lignin) [112], and the fiber-cross-sectional area became the key variable controlling the fiber strength [110]. Aside from that, the extraction method and chemical treatment have an impact on fiber tensile strength. Deka et al. [113] observed that alkali soaking increased the tensile strength of *Parthenium hysterophorus* fiber.

According to De Farias et al. [114], cellulose content has a significant impact on tensile strength and Young’s modulus. The lignin content, on the other hand, had an inverse effect on those strength. Hemicellulose, pectin, and wax, like cellulose, play a role in specific Young’s modulus. The moisture gain of fiber is related to the quantity of hemicellulose and lignin [114]. Microfibril angle (MFA) has a negative and positive relationship with the pectin and lignin content, as well as hemicellulose, respectively, while failure strain value was affected by hemicellulose, lignin, and pectin content, respectively. Cellulose and pectin have a positive effect on density, while wax has a negative effect. Based on this knowledge, it is crucial to investigate the chemical composition of fiber as well as its mechanical and physical qualities.

Glass fiber and natural fibers are extensively distributed, with glass fibers being non-renewable and non-recyclable and natural fibers being the opposite. Natural fibers do not abrade the machine and are not harmful to the lungs when inhaled. According to their disposal viewpoint, glass fiber is a non-biodegradable material, whereas natural fibers are the opposite. Natural fibers have lower tensile strength than synthetic fibers, but they have several advantages, such as not being fractured during processing, equivalent stiffness, and specific strength to glass fibers [115]. They also have less Young’s Modulus and density [116] as well as less energy, density, and cost consumption [117] than synthetic fiber.

To match the use of natural fiber, it was necessary to understand the physical-mechanical properties. Natural fibers have porous qualities, which might make it is difficult to estimate a realistic density. Glass fiber has a higher density of 2.4 g/cm^3^ than natural fiber, which has a density of 1.2–1.6 g/cm^3^. As a result, it can be used to make light-weight composites [58]. The increase in porosity has a proportionate relationship with the lumen size and density of fiber [110]. Fiber bundle diameter tends to rise as density decreases. The specific toughness of bananas, hemp, pineapples, and jute fibers is high. Natural fibers are suitable as reinforcing components in composites because of their unique specific stiffness and tensile strength. Fiber bundle diameter tends to rise as density decreases. The specific toughness of bananas, hemp, pineapples, and jute fibers is high. Natural fibers are suitable as reinforcing components in composites because of their unique specific stiffness and tensile strength. Some of natural fiber, such as jute and sisal fiber, have the potential to replace glass and carbon fibers [118] in composites that demand a high strength-to-weight ratio and weight reduction in that application [119] due to their ease of availability and low cost.

Ramie, sansevieria, pineapple, sisal, and kenaf had low strain (2–6%) but high stress, while *Cocos nucifera* husk fiber had high stress (24%) but low strain [110]. Jute fibers displayed a similar pattern on the stress-strain curve of pineapple fiber [86]. Ramie bast fiber has cellulose content (69–97%) and a low spiral angle (7–12%), as well as a high molecular weight (69–97%), resulting in good mechanical properties [120,121,122,123,124]. Sisal fiber also possesses excellent porosity, tensile strength, bulk, folding strength, and absorbency [90]. Bast fibers were found to have low stiffness, but great strength and elongation, as well as elastic recovery. These fibers are widely available, inexpensive, and function well [125]. The mechanical strength of pseudo-stem banana fiber was also discovered. The flexural and tensile strength of glass fibers were enhanced when sisal or jute fiber were mixed in [31]. The addition of hay fiber, milkweed fiber, kusha grass, and sisal fiber boosted the tensile strength of polypropylene composite [126].

The principal constituents of lignocellulose, which included the cell walls, are cellulose, hemicellulose, and lignin. Aside from that, ash, silica, pectin, waxes, and water-soluble compounds [127,128,129] and oil [128,129] can be found in natural fibers. Plant growth conditions, harvesting period, geographical considerations, and fiber extraction technology all affect the chemical component within the same plant species [127]. To assure the quality of the manufacturing process, the impact of various plant material properties must be analyzed [130]. The fiber characteristic is determined by the angle of the microfibrils and their placement within the cell wall [131]. Figure 3 depicts the position of chemical components in the cell wall (a) and layer position in the secondary cell wall of a plant. Cellulose uses lignin and pectin as glue to join with hemicellulose. The cellulose content of cell wall increases from primary layer (S1) to secondary layer (S2), while the lignin content decreases. Hemicellulose content is found in equal amounts in each layer. The S2 layer is primarily responsible for the physical and mechanical strength of fibers. It has lower microfibrillar angle, higher cellulose content, and contribute to improve fiber strength properties [132,133].

Harvesting method, plant age, sample position in plant, environmental growth condition, and extraction fiber methodology all play role in this diversity, as previously indicated. Cotton linter, cotton, ramie, *Mamordica charantia*, flax, Henequen, and palmyrah are examples of natural fiber with a higher cellulose content than others. Because fibers with low lignin and high cellulose content have high tensile strength, numerous factors might influence this number, and therefore the relationship is not necessarily linear. The crystalline domains of cellulose have a substantial impact on the tensile strength, with more cellulose crystallinity resulting in higher fiber strength. The position of lignin on biomass affects tensile strength because it is wedged between cellulose and hemicellulose.

Some methods have been established in preparation of natural fiber which are summarized in Table 1. Dew retting and water retting process are two common techniques to separate the plant fibers that require about 14 to 28 days to degrade the waxes, pectin, hemicellulose, and lignin [131].

Figure 3 depicts the structures of cellulose, lignin, and hemicellulose [134]. The cellulose content of a plant is determined by its age and species. Cellulose has hexose sugar with greater thermal stability than hemicellulose’s branched structure. It is made up of a linear chain of glucose units linked together by (β (1→4) bonds with a high degree of polymerization (DP). Cellulosic plant fibers have high moisture absorption capacity and poor dimensional stability when exposed to water [144]. The hydrophobicity and hydrophilicity of fibers, as well as their interaction with the matrix, may alter fiber-matrix adhesion with natural fibers as reinforcement [145]. Different cell wall polymers of lignocellulosic materials influence the degradability and properties of natural fibers [146]. Furthermore, cellulose affects natural fiber strength, but lignin prevents UV breakdown and char production [1].

Hemicellulose, which has a low DP, is the third most abundant cell wall constituent of lignocellulosic biomass after cellulose and lignin. As a result, this biopolymer dissolves more quickly in the liquid fraction during biomass pretreatment. Pentose is the most prevalent hemicellulose in non-wood plants, hence a larger concentration effects fiber fibrillation, which raises the bonding potential of pulp sheets [147]. High hemicellulose content improves fiber flexibility during paper sheet usage, allowing it to swell and expand to a large surface area. The higher the crystallinity of cellulose, the less hemicellulose present. As a result, a low hemicellulose content promotes cellulose in the amorphous zone [148]. Thermal, biological, and moisture degradation, as well as absorption, are all caused by hemicelluloses in natural fibers [146]. Hemicellulose material has a favorable relationship with moisture sorption and biodegradation. In addition, hollow natural fibers have a lower/lighter bulk density and contain more water. The degree of crystallinity, orientation, swelling behavior, tensile strength, and porosity of fibers are all affected by their moisture content. As a result, the higher the moisture absorption, the higher the risk of microbial attack [149].

Excess amorphous material such as lignin, pectin, hemicellulose, wax, and cellulose are removed during the alkali process by adding NaOH. Reactive dye fixation is improved by the presence of –OH and –COOH groups in natural fibers [150]. The alkali treatment (adding NaOH) removes excess amorphous content such as cellulose, lignin, hemicellulose, pectin, and wax from natural fibers (*Pennisetum orientale* grass), whereas the alkali treatment removes excess amorphous content such as lignin, pectin, hemicellulose, wax, and cellulose. As a result, the thermal stability and density of the NaOH-treated fibers were higher than those of the untreated fibers and the HCl-treated fibers [151].

Lignin is a biological substance that helps plants maintain their structural integrity [152]. It is also the second most abundant biopolymer with an aromatic molecular structure after cellulose and forms an ester connection with hemicellulose. Lignin molecules include three active functional groups: namely, coniferyl alcohol (G), p-coumaryl alcohol (H), and synapyl alcohol (S). The most frequent connection in lignin is the aryl ether linkage (β-O-4), which accounts for nearly half of all links. This connection is more easily cleaved during lignin conversion and depolymerization. In natural fibers, lignin is also implicated in UV degradation and the formation of char [146]. Coir has a larger microfibrillar angle, as well as a reduced proportion of hemicellulose and cellulose, which impacts plant qualities like strength, durability, damping, wear, weather resistance, and high elongation at break [58].

Since lignin is an unwanted component in raw materials in pulping, it is typically removed during the pulping process in the pulp and paper industry. A complete delignification process could produce pulp with desirable Kappa numbers. Furthermore, delignification of samples with higher lignin content requires a significant quantity of chemical energy. Delignification of samples with decreased lignin content is possible under lower chemical charges and temperatures [147]. Binder-less fiber board with a high lignin content could be used as an adhesive source. It could also be extracted for use in high-value products including adhesives, biosurfactants, antibacterial agents, fine chemicals, lignosulfonate, and so on [1,153,154,155,156,157].

Cellulose content is one of the most important components in assessing the mechanical and physical properties of natural fibers; it is one of three main components (cellulose, hemicellulose, and lignin). The DP was reduced because of excessive chemical treatments such as pulping and bleaching. The number of glucose molecules in one cellulose chain is measured in DP. Depending on the cellulose source, the length of the cellulose polymer chain varies greatly. Plants with a DP of more than 10,000, for example, are vascular cellulose plants. The amount of DP released by a plant is determined by the process used to isolate and treat it. Pure cellulose has a DP of over 10,000 in most cases. Microcrystalline cellulose is another example, which is a high-level crystalline cellulose after going through hydrolyzed acid. Microcrystalline cellulose has a DP value of between 300 and 600 [158,159]. The DP value of microcrystalline cellulose ranges from 300 to 600. This is due to the strong chemical treatment that results in the breakage of the short cellulose chains, and it also influences the crystallinity, mechanical properties, and morphology of cellulose [160,161].

The crystallinity index, or degree of crystallinity, is used to determine the physical and mechanical properties of natural fibers. X-ray diffraction (XRD) at the crystalline peak at 2θ (22.6°) for diffraction intensity I_200_ (crystalline region) and 18° for diffraction intensity I_am_ (amorphous region) can be used to measure the degree of crystallinity of cellulose. The peak height method can be used to calculate the crystallinity [162,163].

## 4. Social and Economic Aspects on Utilization of Natural Fibers

Since prehistoric times, natural fibers have played a vital role in human society as a sustainable and ecologically beneficial source of raw materials that are easily degraded into environmentally friendly items and have the ability to absorb enough moisture. Natural fibers have a variety of fascinating properties, including low density, light weight, low cost, biodegradability, abundant accessibility, minimal health hazards during processing, relatively good basic strength and modulus, good thermal and acoustic insulation characteristics, physical properties, and ease of availability [125,131].

Natural fibers have been favored over synthetic fibers because of their superior qualities [125,131]. Natural fiber has been used as a raw material in a variety of industries, including aerospace, automotive, marine, building and construction, sports and leisure items, electronic appliances, military vehicles, biomedical purposes [10,11,15,16,21,22,164,165,166] as shown in Figure 4: Natural fiber applications are also increasing in textiles, packaging, printed goods, filters, automobiles, furniture, particleboard, insulation board, and other materials [167,168,169,170]. Woven-kenaf aramid and pineapple leaves were used in military vehicles, especially for ballistic purposes [166], and hard armor plate [166,171], respectively. In biomedical applications, natural fibers are in fiber-reinforced composites (FRC), such as various clinical fields [172] as described in Table 2. Hemp and sisal have been reported for utilization as cementitious construction and fancy materials in the construction field [173,174]. In the biomedical field, the most promising natural fiber candidate is undoubtedly cellulose, in the form of nanofibers. Nanocellulose has a variety of biomedical applications, including drug delivery, vascular grafts, skin tissue regeneration, antimicrobial membranes, medical implants, biosensors and diagnostics, and scaffolds [175]. Several methods have been developed to improve the compatibility of natural fibers and polymer matrices in order to enhance the physical and mechanical properties of targeted bioproducts. However, the acceptability of natural fiber and biocomposite materials by the human body is a critical requirement that must be addressed [176].

Natural fibers are a type of biomaterial used for reinforcement of polymer-based composites. Some agricultural plants, including ramie, sisal, and pineapple leaf fiber [107,108,109,110,183] and hybrid fibers of Egyptian and Qatari palm trees [176] and woven cotton fabrics [184] have reported used as bioresources of FRC. The manufacture of natural fiber composite materials or eco-friendly composites has become a popular topic as people become more aware of environmental sustainability. To minimize material weight, natural fibers may be a suitable option for replacing synthetic materials. Natural fiber reinforced polymer and resin composites have been widely used in a variety of industries, including automotive and aviation interior components, as well as military vehicles [166,185,186,187]. Because of their high specific qualities at a lower cost than synthetic fibers, they are appealing for several applications.

Miller [167] mentions the usage of hemp fiber in textile manufacture. The mechanical properties of the bio-based textile composites studied in this review are like those of some traditional materials. The use of pineapple leaf fiber as a reinforcement in the fabrication of yam starch films with packing potential was defined by Mahardika et al. [168]. Asrofi et al. [188] created a bioplastic made of tapioca starch and sugarcane stem fiber for reinforcement. The interior components of an automobile are composed of hemp fiber/polypropylene composites [169], while kenaf and wheat straw were used as vehicle spall-liners and quarter trim panel storage [189]. Natural fiber mats, aluminum sheets, and epoxy resins provide excellent electromagnetic interference prevention while keeping high mechanical qualities in hybrid composites [170]. Good specific properties, low cost, low density, good formability and processability, good mechanical properties, and a plentiful and sustainable source of raw materials are all the benefits of using natural fibers over synthetic fibers. Natural fibers, on the other hand, have a high moisture sensitivity [190,191]. The development of natural fiber composites in a variety of applications has paved new avenues in both academia and industry for the future applications of sustainable natural fibers.

As previously stated, several of the shortcomings of natural fibers should be addressed during the optimization of natural fiber applications. When used as a composite, the hydrophilic nature of natural fibers makes it difficult to adhere to a hydrophobic matrix, resulting in poor mechanical characteristics and processability [112]. Surface treatment methods applied include chemical and enzymatic treatments, corona treatment, and coupling agent addition [119,120,192,193,194,195,196,197]. Furthermore, the handling of the interfacial region before processing with thermoplastics at a temperature up to 200 °C, the interfacial treatment (surface treatment resins, additives, and coating) must be reinforced to address the low degradation temperature of natural fibers [107,198].

The Indonesian government has taken steps to encourage the use of natural fibers, such as appointing an institution to focus on the development of natural fibers and establishing a multi-stakeholder research community, namely, the Indonesian Ramie Consortium (KORI), to study specific natural fibers, primarily ramie. Ramie is a type of natural fiber that has become a national priority in Indonesia for widespread use. Figure 5 depicts the strategy for manufacturing of ramie development in Indonesia.

Manufacturing integration strategies in ramie processing systems to support large-scale production are to be developed with an emphasis on three main sub-systems: cultivation, fiber processing technology, and machining. The ramie-based industry will be able to support the functional value of the fiber or fabric of ramie for functional enhancement of the products. Strategies in business concepts and supply of human resources with competence in all aspects of processing systems will be able to support the realization of the manufacturing of ramie production in Indonesia. This research strategy is currently supported by the Indonesian government in the National Research Program, for the period 2020–2024.

Additionally, the Indonesia Natural Fiber Council (DSI) was founded in Indonesia to assist scientists, policymakers, and other stakeholders in the development of bioproducts generated from natural fibers. DSI proposed a road map for the Indonesian fiber sector from 2020 to 2024, with abaca, kenaf, bamboo, pineapple, sisal, cotton, and ramie as types of promising fiber to be further developed [199]. Furthermore, biduri (*Calotropis gigantea*) is a natural fiber that has the potential to be developed in Indonesia as a thermal and acoustic insulation material and filler material [200] and for winter jacket [201]. Biduri fiber production is predicted to be around 3.6 tons per hectare per year [200]. Some bioproducts, such as biopellets, food, textiles, biocomposites, and ecofriendly shoes, have been launched into the Indonesian market as a result of continued efforts. In addition, several small local businesses extract fiber from fresh pineapple leaves using basic techniques such as retting followed by decortication for clotes, handycraft, and other items. However, the process output is still low, with 2.5 kg of air-dry pineapple fiber produced from 100 kg of fresh leaf fiber and 97.5% of decorticator waste that has yet to be used (visualized in Figure 6). Banana stems are treated in a similar way to make banana fibers in this local enterprise. Until now, cotton has been the main fiber source in the Indonesian textile industry, but the qualities of local cotton have not met the requirements, so nearly all of it is imported, while the other fibers have been used to their full potential. Considering the potency and challenge, continual efforts to disseminate information about the various uses of natural fibers in the community are required.

Natural fibers play an important role in improving the quality of human life. However, waste can be generated during the product life cycle and during the processing of natural fibers into bioproducts. To achieve the most efficient utilization of resources, waste management should be conducted continuously by recycling and/or upcycling of waste, aside from innovation in the design of bioproducts. Shanmugam et al. [202] recognized recycling and the use of bio-based constituents as essential issues in adopting a circular economy (CE). CE adheres to the principles of reduce, reuse, recycle, and replace. CE is beneficial to the environment, economy, and society when used in FRC manufacturing. Given the numerous sustainability challenges confronting our societies, transitioning to a circular economy and closing resource loops through recycling is a viable solution [203]. Figure 7 proposes a CE concept based on natural fibers that is more considered than a linear economy concept for future resource conservation and environmental balance. The CE approach is gaining traction and has been proposed in some fields, such as carbon fiber manufacturing [204], agricultural sector [205] and biomass biorefinery [206], for gradually reducing energy consumption during the manufacturing process. Biomaterials in the CE present numerous challenges for the industry in terms of developing new network and commercial opportunities while remaining focused on consumer demands [206].

The development of an information system for Indonesian natural fibers, as well as collaboration with a variety of stakeholders such as research and development institutions, industries, policymakers (local and national), and universities, are ongoing efforts to bring Indonesian local industry independence. National innovation products made from natural fibers can be created by local industry in the future and sold at least locally, with Indonesians consuming them.

Natural fiber as lignocellulosic biomass has an economic chance to meet industrial needs, depending on the processing level that has been made to make its derivative products, including its market to accomplish. According to Ruamsook and Evelyn [207], there are four levels in which biomass can be processed and turned into value-added goods before being sold to potential demand markets (Figure 8). Farmers become the first important people actors to create their biomass as the major components of industrial needs, as indicated in this picture. Commodities such as corn, wheat, cotton, and hay, as well as other crop farms such as paddy, are the possible resources of rubber and polymer markets. Many industrial polymers and plastics are still made from non-renewable oil and gas resources today. This would cause a supply shock when non-renewable resources are depleted, causing the processed product to bubble to an unacceptably high price [207]. As a result, the growing interest in bio-based polymer and plastic products derived from renewable sources creates a market opportunity for biomass in exchange for enhanced environmental support in reducing climate change pressure.

Farmers will benefit from the potential use of paddy waste as an alternative source of packaging because they have been heavily reliant on agriculture without any additional revenue, as most farmers are still subsistence and have a low-middle income. On the other hand, this will contribute to reducing future climate change issues by allowing farmers to benefit from better climate conditions through sustainable agriculture. Nonetheless, the government must promote this innovation to increase economic potential and to provide an instrument for industrial businesses to improve their knowledge of the sustainable industrial environment. It may not be easy, but once the government steps in to regulate the industrial ecosystem by paying more attention to reducing plastic waste and implementing sustainable bioplastic for bio-packaging for both large industrial ecosystems and small-medium enterprises, it could have structural potential. One of the studies that uses paddy-waste as bioplastic is the usage of rice straw cellulose (*Oryza sativa*) as bioplastic by the pulping process and phase inversion method [208].

Building an ecosystem of sustainable industry, particularly for consumer behavior, is to use more sustainable packaging or bio-packaging that can be created from natural fiber. As it is known that the production of food packaging made from plastic as well as styrofoam is about 14.000 tons per year, it has affected the use of plastic packaging for food [203]. However, this material is not environmentally friendly and could cause a significant impact both on the user and the environment after its usage, with a long-term impact on climate change. Thus, the government could develop bio packages as a particular potential both for reducing environmental issues and improving the economic opportunity of farmers from the paddy waste produced. Basically, this novel innovation could be started by a small-scale industry where a group of farmers could start to process the paddy waste materials where generally they would not be sold except for burning.

On the other hand, the development of bio-packages made of paddy straw benefits not only farmers in terms of economic earnings, but also consumers, as they have paid more for the environmentally friendly food package to have both future-health preservation as well as the original flavor and scent of stored food from this bio-package material, compared to a conventional plastic food package, which has more influence on their food, particularly when the foods are still at high temperatures. With these kinds of benefits, coupled with intense regulatory aspects, consumers would indeed be willing to pay more to get this type of food package if the government could guarantee that it would not harm them and be rigid in implementing environmental policy in general. If this is accomplished, the introduction of novel food packages made of paddy straw will be imminent, benefiting farmers who are the primary source of this material. Many countries have taken steps to encourage the use of natural fibers, such as appointing an institution to focus on the development of natural fibers and establishing a multi-stakeholder research community.

## 5. Future Prospects

Many different types of natural fibers are potential raw materials for bioproducts, but it is necessary to select the most locally viable fiber before attempting to employ it in an industry. In several fields, using local potential natural fiber for industrial purposes helps to lessen reliance on imported products. Some efforts will be made to speed up the exploitation of local raw materials for local industry, thereby assisting in the establishment of local industry, mostly on a small and medium scale. The wide diversity of natural fiber characteristics remains a difficulty in manufacturing consistent quality of bioproducts. Therefore, understanding the features is essential when processing natural fiber effectively. Processing of natural fiber with environmentally friendly technology and appropriate procedures should be used in the future as environmental concerns grow for the preservation of sustainable nature. Kenaf and ramie, bamboo, bananas, and pineapple have all been used as commercial bioproducts for a variety of industries, including automotive, building materials, handicrafts, and textiles. Up to now, there has been no exposed intermediate industry in natural fiber; therefore, the biocomponent industry has been initiated in recent years and is able to create environmentally friendly footwear products. This footwear can be built with biocomponents from natural bioresources such as natural fiber. In summary, viability, suitable technology, and social issues in the development of natural fibers are significant components that, when controlled together, can stimulate the use of bioproducts made from natural fibers. It is critical that respective ministries should be involved in natural fiber development and prioritize the agenda to ensure a strong supply chain and sustainability of natural fiber production. The Ministry of Agriculture should be in charge of providing sufficient land for planting as well as the necessary technology for a successful harvest. While the Ministry of Industry can provide the manufacturing technology and create the product diversification design.

## 6. Conclusions

Natural fibers with enticing properties such as lower density, lighter weight, biodegradability, good specific strength and modulus, good thermal insulation, good acoustic insulation, and high electrical resistance can be used for various applications. Furniture, automotive (car bumper beam, disc brake pads), electronic industries (automobile components), and building construction (molded panel components, window frames) are only a few of the applications for natural fibers. Biomedical application refers to the potential use of natural fibers to cover tissue engineering, biomedical implants, and drug delivery systems that must be biocompatible with the human body. Although natural fibers such as bananas, abaca, pineapples, bamboo, cotton, ramie, sisal, coconut, sansevieria, jute, and kenaf are plentiful and readily available, industrial applications require access to the most readily available. Ramie is being used as a model for developing a national priority bioproduct based on textile innovation, which is facilitated through government support. This way, it will coordinate the efforts of all stakeholders, including industry, research and development agencies, and farmers, in order to maximize benefits. A circular economy that is sustainable can be achieved by developing a biocomponent industry for bioproducts. The establishment of an information system for natural fibers, as well as collaboration with a variety of stakeholders such as research and development institutions, companies, policymakers (local and national), and universities, are ongoing efforts to increase the use of natural fibers in a sustainable circular economy.

## Figures and Tables

**Figure 1 polymers-13-04280-f001:**
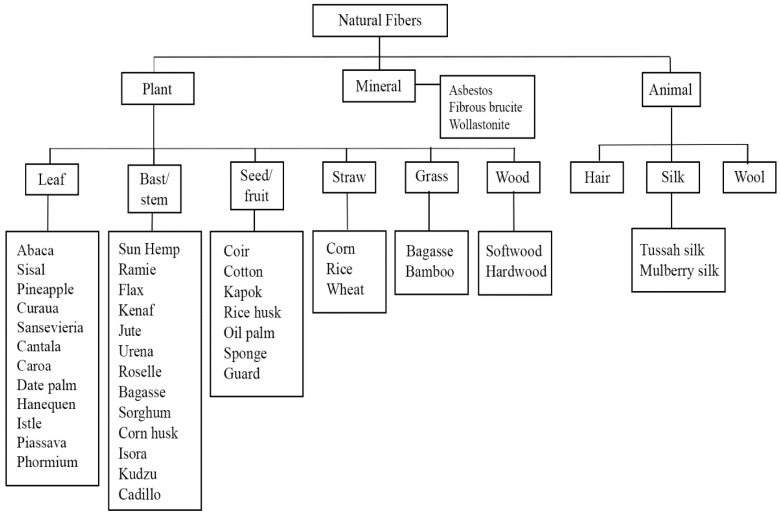
Schematic representation of fiber classification, reprint with permission from ref. [26]. Copyright © 2021 Woodhead Publishing Limited.

**Figure 2 polymers-13-04280-f002:**
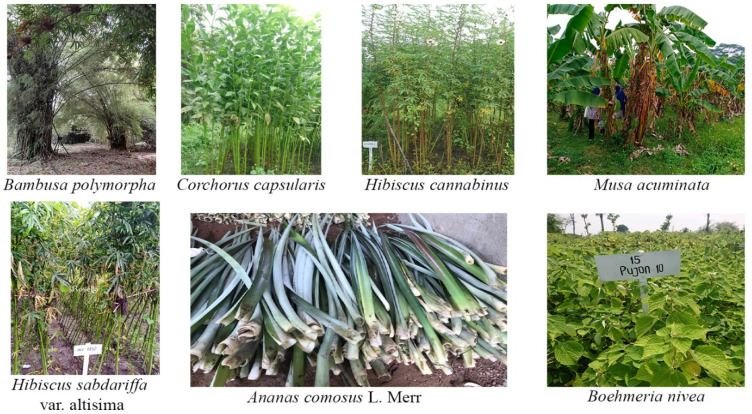
Some natural fiber resources (Pic courtesy authors collection).

**Figure 3 polymers-13-04280-f003:**
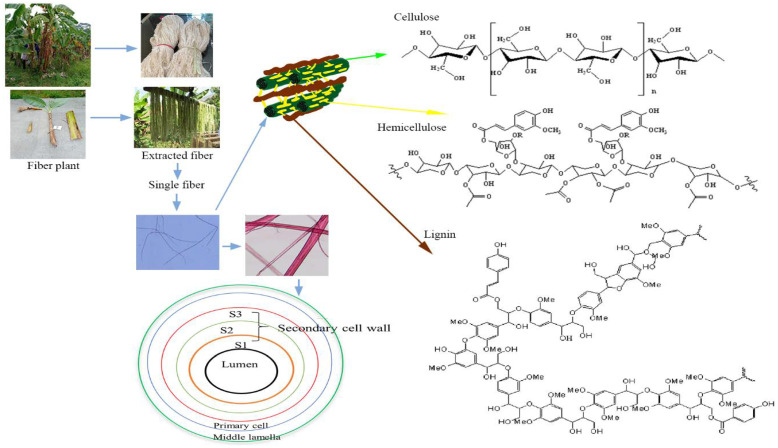
The hierarchical cell wall structure of lignocellulosic biomass (adapted from [134], Copyright © 2021 Elsevier Inc., License Number: 5193031271486) Cell wall layer position of plants (modified from [135], Copyright © 2021 Elsevier Ltd., License Number: 5193040344374).

**Figure 4 polymers-13-04280-f004:**
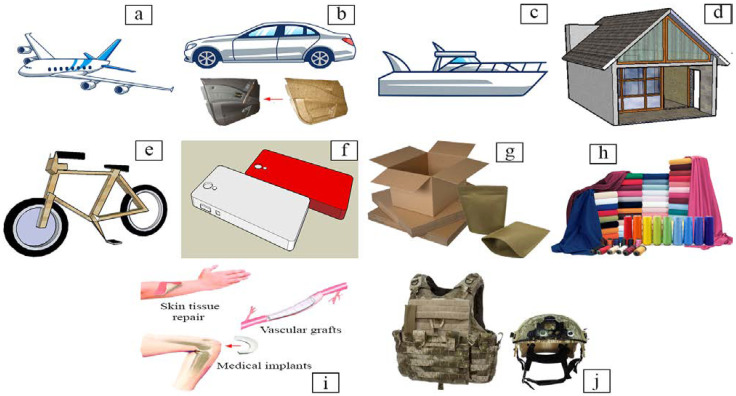
Potential application of natural fibers in many sectors such as (**a**) aerospace, (**b**) automotive, (**c**) marine such as boat hulls, (**d**) building and construction such as insolation board, (**e**) sport and leisure goods, (**f**) electronic appliances such as handphone casing, (**g**) paper and packaging, (**h**) textile, (**i**) biomedical, and (**j**) military fields.

**Figure 5 polymers-13-04280-f005:**
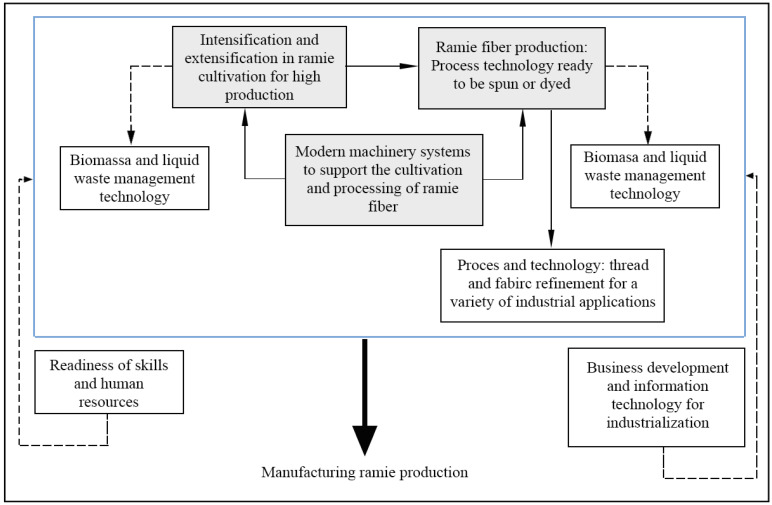
Manufacturing integration strategy in the ramie production system in Indonesia.

**Figure 6 polymers-13-04280-f006:**
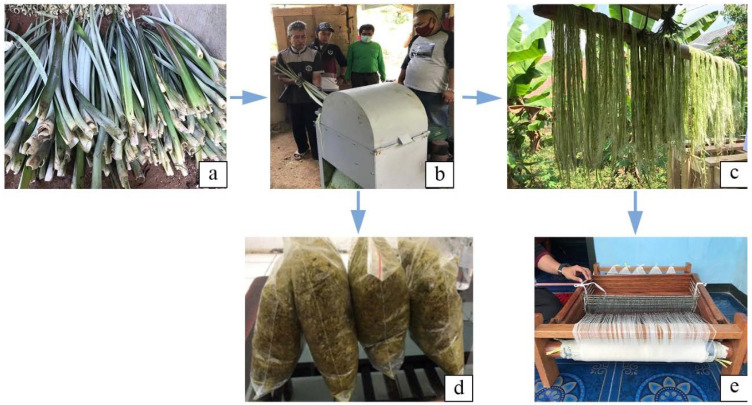
The process production of extracted fiber from fresh leaf pineapple, (**a**) fresh leaf fiber, (**b**) decorticator process, (**c**) wet extracted fiber, (**d**) decorticator waste, (**e**) dried fiber that ready for spinning.

**Figure 7 polymers-13-04280-f007:**
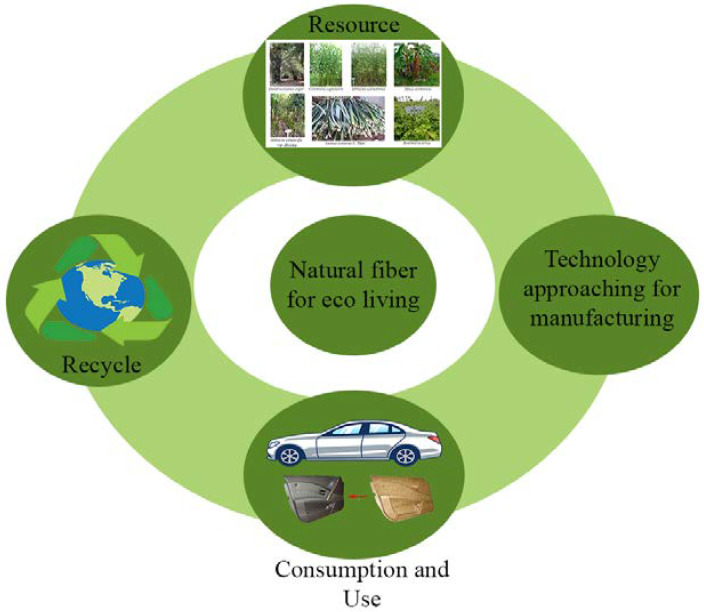
Circular economy concept to conserve the natural fiber for future eco living.

**Figure 8 polymers-13-04280-f008:**
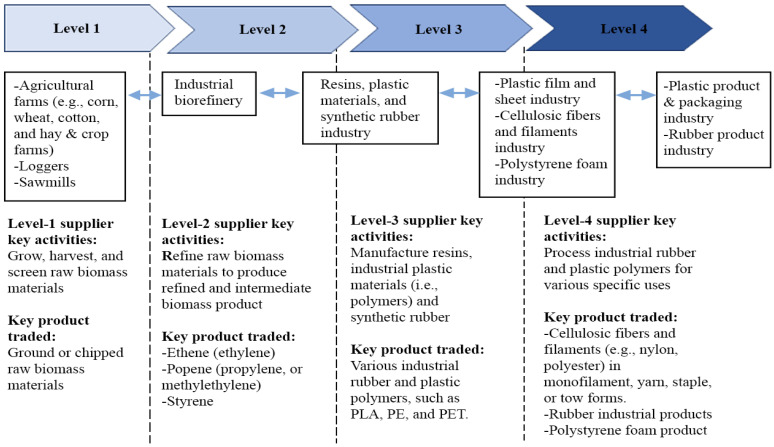
Simplified example of multi-level rubber and polymer markets for biomass (adapted from [207]).

**Table 1 polymers-13-04280-t001:** Preparation method for producing the extracted natural bast fibers.

Introducing Methods	Advantages	Disadvantages	References
Dew retting	Relative ecofriendly or less energy process by using bacteria and moisture in the plant for separating individual fibers from the plant Common in areas with heavy night dew and warm days, as well as areas with water shortages.	Excessive retting brings difficulties in separating individual fibers or tend to weak the fiber strength Required long processing time (2–3 weeks) depending on climatic condition Obtaining dark fiber and poor quality	[15,136]
Water retting	Produce fiber with high cellulose content, which gives the fiber a higher tensile strength Produce fiber with lower density which is suitable for low weight composite applications Duration of process for 7–14 days	Need surface treatment as initial step to increase the surface roughness High cost High water treatment maintenance	[17,137,138,139,140,141,142,143]
Mechanical extraction	Produce a significant amount of acceptable quality of fibers Short time process duration	Damage fiber cell wall structures, resulting in dislocations, kink bands or node that have a negative impact on tensile mechanical properties and may even compromise composite performance High cost	[15,134,141]
Chemical treatment	Produce fiber with high cellulose content, higher tensile strength, thermal stability, and crystallinity index The surface roughness of the fiber relatively good (based on SEM analysis) Enhance the physicochemical properties of the fibers	Some chemical treatment waste can pollute the environment	[17,137,138,139,140,141,142]

**Table 2 polymers-13-04280-t002:** Application of natural fiber composites in biomedical field.

Specific Area Application of Fiber Composite	Source of Fiber	References
Blood bag Drug/gene delivery scaffold	Pineapple, rambutan and banana skin	[177]
Ancient medicine Modern functional food	Flax and flaxseed oil	[178]
Wound dressings	Flax	[179]
Drug delivery	Cotton	[179]
Wound healing	*Bombyx mori* silk	[180]
Tissue engineering Drug delivery Wound dressing Medical implants Cardiovascular implants Scaffolds for tissue engineering	Pineapple leaf	[181,182]
Prosthodontics Orthopedics Cosmetic orthodontics	-	[176]
Dental application	-	[20,21,22]

## Data Availability

The data presented in this study are available on request from the corresponding author.

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
