# Peer review of "A Comprehensive Review on Natural Fibers: Technological and Socio-Economical Aspects"

_polymers, 2021, doi:10.3390/polym13244280_

Round 1

Reviewer 1 Report

The current topic on natural fibers is timely and important but the manuscript lacks the scientific rigour required for a journal such as Polymers. I have several concerns with the manuscript as follows.

Title: The title needs to be revised as per the manuscript. The use of the word "potential" is incorrect for a review.

Figure 3: Have the authors sought permission for using/adapting the figures from the relevant sources. It should be checked for all the figures where applicable.

Figure 4: The pictures could be in vector format rather than actual images.

Sections 2 and 3 are very tedious and do not provide information directly related to the topic reviewed.

Section 4: Biomedical applications of these fibers are missing and need to be provided in detail.

Section 4: Very surprising to see no mention and detailing of circular economy?

Author Response

Dear valuable reviewer

Thank you very much for your suggestion on our manuscript. We tried to respond to your comment the best we can. the revised part has been presented in green highlight

Reviewer 1:

The current topic on natural fibers is timely and important but the manuscript lacks the scientific

rigour required for a journal such as Polymers. I have several concerns with the manuscript as

follows.

  1. Title: The title needs to be revised as per the manuscript. The use of the word "potential" is incorrect

for a review.

Author’s response:

The title has been modified by removing the “potential” from the title

  1. Figure 3: Have the authors sought permission for using/adapt the figures from the relevant

sources. It should be checked for all the figures where applicable.

Author’s response:

The authors have been received permission from Elsevier as the publisher of the article through RightsLink with license number 5193031271486 for Bertella and Luterbacher, 2020 and 5193040344374 for Kabir et al., 2012.

  1. Figure 4: The pictures could be in vector format rather than actual images.

Author’s response:

Figure 4 has been revised as suggested with additional Figures 4i (biomedical application) and 4j (military application)

  1. Sections 2 and 3 are very tedious and do not provide information directly related to the topic

Author’s response:

As the topic that becomes concerned in this manuscript is to overview the technology and socio-economic aspect of natural fiber thus in sections 2 and 3 in the author’s opinion is required to lead to a more comprehensive knowledge of natural fiber such as classification and potency some kinds of natural fibers. In section 3, to bring the technology processing by evaluating the natural fiber properties as considering appropriate treatment.

  1. Section 4: Biomedical applications of these fibers are missing and need to be provided in detail.

Author’s response:

Biomedical application has been added with an additional table (Table 2) and also discussion in section 3 (green highlight) on page 17

  1. Section 4: Very surprising to see no mention and detailing of circular economy?

Author’s response:

The additional discussion of the circular economy concept has been added in section 4 (page 21, green highlight)

Best regards

Dr. Widya Fatriasari

Reviewer 2 Report

Please find the enclosed comments word file. Thanks 

Author Response

Dear reviewer 

I would like to thank four of your valuable time and significant suggestions to improve our manuscript. The revision part has been presented in green highlight. For your detailed concern, we have tried to respond the best we can do that can be checked as follow.

Reviewer 2

Reviewers’ comments

Comments

The review under the title “A comprehensive Review on Potential Natural Fibers: Technological and Socio-Economical Aspects” It is an interesting topic, and I see, the authors have addressed this topic thoughtfully in a scientific point of view. However, the abstract and conclusions presented here are well understood in open literature with respect to all the information presented here. Please highlight the new knowledge or unique updates that has been an injected by authors. As whole the review has a good description, but the figures and tables are not in good format which I mention below in the detail.

Author’s response:

Thank you very much for your valuable comment on the manuscript for improving the manuscript quality. As per your suggestion, we have revised the abstract and conclusion by emphasizing the highlight of the review analysis. The changes can be found in the green highlight.

The Figures have been revised in better format form (please check all) by redrawing them

  1. Page 3, figure 1. Schematic representation of fiber classification. The horizontal lettering inboxes is perilously small and hard to read. At a minimum, use larger fonts for all.

Author’s response:

Figure 1 has been redrawn with larger fonts showing more clearly the figure and present in horizontal layout

  1. Page 6, figure 2. Some natural fiber resources. I can see the English name of plants but I would suggest, but the scientific name of plants as well for better understanding.

Author’s response

The scientific name of plants in Figure 2 has been added to each plant as follow

Ramie (Boehmeria nivea); Kenaf (Hibiscus cannabinus); Rosela (Hibiscus sabdariffa var. altisima); Yute (Corchorus capsularis). Pineapple (Ananas comosus L. Merr) leaves, Bamboo (Bambusa polymorpha), Banana (Musa acuminata)

  1. Figure 3, page 8. The hierarchical cell wall structure of lignocellulosic biomass. Make fonts in the graphic large because the chemical structure of Cellulose, hemicellulose, and lignin are not readable.

Author’s response:

Figure 3 has been revised in a more clear picture with a structural component of fiber has been enlarged

  1. Table 1, page 9. I can see the columns inconsistency, Advantages, and Disadvantages lines bring into Align to left or make it center text but should be consistent.

Author’s response:

Table 1 has been revised into a more consistent way for each column and raw

  1. Figure 5, page 13. The boxes have an inconsistent font, which is completely different fonts than the rest figures and tables. Use the same font style and size for all figures and tables in review.

Author’s response:

Figure 5 has been revised to present uniform in the font type (times new roman) and size with table

  1. Figure 7, page 15, Boxes lettering are small, use larger fonts.

Author’s response:

Figure 7 has been revised in the larger font in the box

  1. Page 4, line 153, the authors should not format “sp.” In italic. Check throughout the entire review.

Author’s response:

The "sp" has been revised in the not italic format and has been checking the entire manuscript in a similar way presentation

  1. In my understanding, the authors should use same format “Ahmed et al. (2019).” Page 12, line no, 474 showing different format. Check throughout the entire review and make it all at same style.

Author’s response:

The format “Ahmed et al. (2019)” has been revised and has been checking the entire manuscript in the similar format

  1. The review to have improvements to format consistency.

Author’s response:

The format of the manuscript has been checked the consistency

  • Make the list of Acronyms throughout the entire review

Author’s response:

The acronym has been prepared and put in the last manuscript

  • References are not in good shape, I would suggest using EndNote, Zotero, etc. to make all references in the same format and bring them into better shape. For instance, reference no. 181, 812, etc.

Author’s response:

The references have been reformatted by using endnote reference manager to present a better and consistent presentation all of the references

Thank you very much

Best Regards 

Dr. Widya Fatriasari

Round 2

Reviewer 1 Report

No further comments.